# Seroprevalence of the *Strongyloides stercoralis* Infection in Humans from Yungas Rainforest and Gran Chaco Region from Argentina and Bolivia

**DOI:** 10.3390/pathogens9050394

**Published:** 2020-05-20

**Authors:** Rubén O. Cimino, Pedro Fleitas, Mariana Fernández, Adriana Echazú, Marisa Juarez, Noelia Floridia-Yapur, Pamela Cajal, Alfredo Seijo, Marcelo Abril, Diego Weinberg, Pablo Piorno, Nicolás Caro, Paola Vargas, José Gil, Favio Crudo, Alejandro Krolewiecki

**Affiliations:** 1Instituto de Investigaciones de Enfermedades Tropicales (IIET-CONICET), Sede Regional Orán, Universidad Nacional de Salta, Orán 4530, Argentina; pedro.fleittas@gmail.com (P.F.); adrianaechazu@hotmail.com (A.E.); marjua_78@hotmail.com (M.J.); narfy89@gmail.com (N.F.-Y.); spcajal@yahoo.com.ar (P.C.); nicolascaro2010@hotmail.com (N.C.); paola.a.vargas.f@gmail.com (P.V.); jgil@conicet.gov.ar (J.G.); alekrol@hotmail.com (A.K.); 2Cátedra de Química Biológica, Facultad de Ciencias Naturales, Universidad Nacional de Salta, Salta 4400, Argentina; 3Asociación para el desarrollo sanitario regional (ADESAR), San Antonio de Areco, Buenos Aires 2760, Argentina; marianaf411@gmail.com (M.F.); laboratorio@murzicatopiorno.com.ar (P.P.); fgc1937@yahoo.com.ar (F.C.); 4Instituto de Patología Experimental, Facultad de Ciencias de la Salud (IPE-CONICET), Universidad Nacional de Salta, Salta 4400, Argentina; 5Hospital Muñiz, Buenos Aires, CABA C1282AEN, Argentina; ceijo@intramed.net.ar; 6Fundación Mundo Sano, Buenos Aires, CABA C1061ABC, Argentina; mabril@mundosano.org (M.A.); dweinberg@mundosano.org (D.W.); 7Instituto de Investigaciones en Energía no Convencional—Sector Salud y Ambiente (INENCO-CONICET), Universidad Nacional de Salta, Salta 4400, Argentina

**Keywords:** *Strongyloides stercoralis*, seroprevalence, epidemiology, Argentina

## Abstract

The threadworm, *Strongyloides stercoralis*, is endemic in tropical and subtropical areas. Data on the prevalence and distribution of infection with this parasite species is scarce in many critical regions. We conducted a seroprevalence study of *S. stercoralis* infection in 13 locations in the Gran Chaco and Yungas regions of Argentina and Bolivia during the period 2010–2016. A total of 2803 human serum samples were analyzed by ELISA-NIE which has a sensitivity of 75% and specificity of 95%. Results showed that 551 (19.6%) of those samples were positive. The adjusted prevalence was 20.9%, (95% confidence interval (CI) 19.4–22.4%). The distribution of cases was similar between females and males with an increase of prevalence with age. The prevalence in the different locations ranged from 7.75% in Pampa del Indio to 44.55% in Santa Victoria Este in the triple border between Argentina, Bolivia, and Paraguay in the Chaco region. Our results show that *S. stercoralis* is highly prevalent in the Chaco and Yungas regions, which should prompt prospective surveys to confirm our findings and the design and deployment of control measures.

## 1. Introduction

*Strongyloides stercoralis* is a nematode of medical importance widely distributed all over the world affecting several hundred million individuals [1,2]. This threadworm intestinal parasite that infects dogs, cats, and primates including humans is endemic in tropical and subtropical regions with poor sanitation conditions. The infection is frequently asymptomatic and can persist for years without detection [3]. 

The Gran Chaco is a warm subarid region of 1 million km^2^ representing the second largest biome in the Americas after the Amazon region, crossed by the Tropic of Capricorn, hosting almost 10 million people in Bolivia, Paraguay, Brazil, and Argentina. With low average population density, it has been identified as a “hot spot” for Neglected Tropical Diseases (NTDs) that requires special emphasis for disease control. Chagas disease and soil-transmitted helminthiasis (STH), including *S. stercoralis* are the main NTDs with active transmission in the region although burden information is incomplete [4,5,6]. Subtropical Yungas are distributed in northwestern Argentina and southern Bolivia over approximately 56,000 km^2^ and represent the austral limit of the wooded system known as the Andean Yungeño forest extending from Venezuela to Argentina. This vegetation type expands across a large altitudinal gradient (400–2300 masl), where tree species turnover promotes the occurrence of three altitudinal belts: (i) pre-montane (400–900 m asl), (ii) lower forest (900–1600 m asl), and (iii) upper montane forest (1600–2300 masl) [7]. 

Argentina is among the countries of Latin America endemic for STH although with varying degrees of prevalence. The areas of high prevalence in Argentina were found in the provinces of Misiones, Chaco, Formosa, and Salta, all of them in the northern of the country [8,9,10]. 

*S. stercoralis* is an exception among soil transmitted helminths of medical importance because it can reproduce within the human host (autoinfection cycle) and allows the infection to perpetuate as a chronic state, which can last for decades. The clinical presentation is varied, and depends on the intensity of the infection and immunological states of the individual. Most patients are asymptomatic, while typical symptoms are abdominal pain, diarrhea, and urticaria [2,3]. The disseminated form of the infection, or hyperinfection syndrome, is most frequently seen in immunosuppressed patients (e.g., transplant recipients, HIV or HTLV-1 infections, corticosteroid use) who experience a life threatening complication triggered by an exponential increase in larvae production and migration to extraintestinal sites [1]. Typically, strongyloidiasis is contracted by the skin penetration of the infective larva (L3) from contaminated soil. The eggs produced by the adult female worm located in the small intestine and the larvae are released in stools. The treatment of choice for strongyloidiasis is ivermectin [2].

To date, most STH prevalence studies are carried out using egg counting methods (Kato-Katz, MiniFLOTAC and McMaster´s), whereas techniques like Baermann, Agar plate, and sedimentation/concentration (Telemann) are designed for the detection of larvae of *S. stercoralis* in stools. However, these techniques are complex and have a relatively low sensitivity [2]. Recent innovations like qPCR, although superior in several reports have not shown significant superior sensitivity in a recent systematic review [11]. Serology has been used in a growing number of surveys appearing as a useful tool for prevalence estimations of *S. stercoralis* [12,13,14,15,16,17].

Serological methods are more sensitive and practical than the examination of stools. A variety of commercial kits and in-house tests using either crude or recombinant antigens have been used with different techniques, such as ELISA, IFAT, Luminex, and LIPS for the diagnosis of *S. stercoralis* infections [18,19]. The sensitivity of these serological assays varies from 70% to 100%, while the specificity is improved when recombinant or purified antigens are used instead of crude antigens [20,21,22,23]. The NIE recombinant antigen, a 31-kDa antigen derived from *S. stercoralis* L3 parasites, represents an alternative for serological diagnosis, with reported sensitivities and specificities of 84–98% and 95–100%, respectively, being comparable in performance to the crude antigen-based ELISA [19,23,24,25,26,27,28].

The purpose of this study was to report the seroprevalence of *S. stercoralis* infection in a wide region of the Gran Chaco and Yungas regions in northern Argentina and southern Bolivia using NIE-ELISA, in order to contribute to the understanding of the burden of this infection in the region.

## 2. Results

### 2.1. Characteristics of the Populations

A total of 2803 serum samples from 13 different rural, urban and peri-urban localities distributed through the Gran Chaco and Yungas region from Argentina (12 localities) and Bolivia (one locality) were included in the analysis (Table 1). Of the 2218 samples with demographic information available (excluded Pampa Indio), 42.3% (*n* = 939) were male and 57.7% (*n* = 1279) were female. The age ranged from 1 to 92 years, with a median age 12 ± IQR: 21.25 years old. Among the different age groups, 10.5% (*n* = 228) were <5 years old, 45.9% between 5 and 15 years old (*n* = 996) age years old, 31.6% between 15 and 45 years old (*n* = 686), and 11.9% (*n* = 259) ≥ 45 years old.

### 2.2. Seroprevalence Rates

The overall *S. stercoralis* seroprevalence was 19.6% (95%CI: 18.2–21.1%) (551/2803) while the adjusted seroprevalence was 20.9% (95%CI: 19.4–22.4%). A wide range of prevalence rates between communities was observed with the lowest (7.75%) in Pampa del Indio and the highest (44.55%) in Santa Victoria Este a rural locality at the triple-frontier of Argentina, Bolivia, and Paraguay in the Gran Chaco (Figure 1 and Figure 2). Overall, there was no significant difference in prevalence rates between females and males (22.4% and 21.7%, respectively) (*p* = 0.760), with just one locality (SRN Oran) where the difference was significant (*p* < 0.001). The comparison in the prevalence rates between the different age groups with respect to the age group <5, showed a significant increase with respect to the age groups 15–44 and ≥45 years old (*p* < 0.001). No significant difference was observed with the age group 5–14 years old (*p* = 0.38) (Figure 3).

## 3. Discussion

With over 2000 samples from 13 different communities, this serological survey is the largest report on the current situation of *S. stercoralis* prevalence in the Chaco-Yungas region of the Americas, including communities from diverse localities across the region, highlighting the significant presence of this STH in the population living in this region as well as the need for surveys assessing the burden and distribution of NTDs in the impoverished communities that occupy this area.

Although methods such as Baermann or Agar plate are considered adequate for the diagnosis of *S. stercoralis* infections, their sensitivity is significantly lower than serum antibody detection methods although improved when multiple samples rather than a single stool are analyzed [2,29]. Unfortunately, the use of the Baermann or Agar plate methods is not common in the laboratories of the local health institutions and these methods are rarely used for epidemiological studies. Serological techniques such as ELISA have the advantage that large quantities of samples can be analyzed in a reasonable time and at lower costs [21,30]. Previous reports on the use of the NIE-ELISA technique in epidemiological surveys in an endemic region and in populations where mass drug administration (MDA) campaigns are conducted [13,16,31], have demonstrated the validity and feasibility of this approach that allows the possibility of processing large quantities of samples while also integrating it with other sero-surveillances [17,23,32].

For the current analysis we defined the sensitivity and specificity parameters in 75% and 95% based on a blinded comparison between different serological methods and performed the prevalence adjustment based on those parameters [27,33,34]. 

The results obtained in this study highlight the presence of *S. stercoralis* in the Chaco region and its neighboring Yungas region and confirms previous smaller and more localized reports, which informed prevalence ranging from <5% to over 50% [9,35,36,37,38]. Only two localities in our study presented a prevalence < 10%, Isla de Cañas (8.7%) and Pampa del Indio (7.8%). Systematic reviews on the prevalence of *S. stercoralis* in the region agree on the scarce information and the need for more surveys with adequate design, including standardized, reproducible, and accurate diagnostic methods although still identifying areas with a prevalence > 20% [8,9]

Several recent studies in South America and Asia identified areas with overall prevalence of *S. stercoralis* infections around 20–25%. This prevalence can increase in rural populations without access to drinking water and sanitary systems [16,39,40,41,42]. In Bolivia, a recent study estimated a global seroprevalence of 22% in Cochabamba Department and 24% in Santa Cruz Department [34]. In the same study, prevalence measured by parasitologic methods averaged 7%, demonstrating that the use of direct methods in epidemiological studies leads to an underestimation of the prevalence of *S. stercoralis* infection. In our study, the seroprevalence in Gutierrez (Department of Santa Cruz) was 20.7%, similar to that previously reported in Santa Cruz Department. The population studied corresponds to rural communities, with dirt floors, no electricity, and without improved water and sanitation facilities [34,43]. Our results and conclusions are however somewhat different than those described in a sero-epidemiological study showing a decreasing trend in the prevalence in the Chaco region through a comparison of surveys performed in 1987 (122 samples) in the communities of Camiri and Javillo (both in Santa Cruz Department) and 2013 (233 samples) in Bartolo (Chuquisaca Department) and Ivamirapinta (Santa Cruz Department) [44]. In that report the seroprevalence drops from 16% to 6%, although geographic differences within the Chaco region, which includes humid rainforest and dry foothills, place limitations in longitudinal comparisons using different communities (Figure 3). Additionally, that study found a higher prevalence at younger ages; in this study we observe that the prevalence increases according to the age group, which coincides with that reported by Steinmann et al. [29] and Schär et al. [45] in rural villages of China and Cambodia, respectively. A similar age distribution was reported in a sero-epidemiological study in the Peruvian Amazon, which calculated that the odd ratios according to age increases between 1 or 2 points per year of age [12].

There is evidence that variables related to personal hygiene such as the presence of a latrine in the home, frequent use of shoes, use of slippers to go to the latrine, and handwashing decrease the risk for *S. stercoralis* infection [42,46]; in our report it was not possible to make any inference about these behavioral and socioeconomic factors, although in a previous study from our group in communities in the Chaco region of Argentina, there was an association between the lack of adequate sanitation and the presence of infection with hookworms and/or *S. stercoralis* [14].

A limitation of this study is that we were not able to determine the active infection status at the time of sampling, since the detection of anti-NIE antibodies does not allow to determine if the infection is present or passed, since the drop in antibody titers measured through anti-NIE antibodies can take several months [13,25]. There were however, no deworming activities with drugs active against *S. stercoralis* in the communities included in this study, prior to the surveys [43,44].

## 4. Materials and Methods 

### 4.1. Study Design

Different community based serologic surveys on humans were carried out from 2010 until 2016 in different localities of the Gran Chaco and Yungas regions with the goal of evaluating the prevalence and distribution of different NTDs (Table 1 and Table 2). Sampled individuals were randomly selected from a population-based census using either individuals or households as the units for randomization (Table 2). These studies were carried out in 12 localities from Argentina from two different provinces (Salta and Chaco) and one from Bolivia (Gutierrez Municipality, Santa Cruz Department) (Figure 2). The surveys were part of a wider project aiming to define strategies for STH control. MDA campaigns with single doses of ivermectin (200 µg/kg) and albendazole (400 mg) free of charge to all participants from the localities (except Las Leonas, Pampa del Indio, and Gutierrez) was offered after the collection of serum samples.

### 4.2. Serologic Assay

Seroprevalence was measured using the recombinant antigen NIE from *S. stercoralis* [24]. The enzyme-linked immunosorbent assay with NIE (NIE-ELISA) was performed as has been previously described with minor modifications [13]. A standard curve was used and values (Unit/mL) interpolated from that standard curve. Cut off was defined using negatives and positives controls sera from stool positive *S. stercoralis*-infected patients and healthy non-infected individuals. The cut off was set at 120 Units/mL corresponding to sensitivity and specificity of 75% and 95%, respectively. All blood specimens were centrifuged, and the serum was separated and frozen at −20 °C until examination by NIE-ELISA for detection of anti-NIE IgG. 

### 4.3. Statistical Analyses

All data (result of serology, age, sex, and location) were organized in Microsoft Excel tables. The adjusted prevalence of *S. stercoralis* infection, including the 95% confidence intervals (95%CI), was calculated using WinEPI: Working in epidemiology (Blas I, 2006; Universidad de Zaragoza, http://www.winepi.net/menu1.php) following formula: AP = (OP + Sp - 1)/(Se + Sp - 1), where AP is the adjusted prevalence, OP is the observed prevalence of positive serological test result found in our study, and Se (75%) and Sp (95%) are the sensitivity and specificity estimates found in the study cited earlier [34]. For the analysis of prevalence according to sex and age, data from Pampa del Indio was excluded because that information was not available.

### 4.4. Ethical Considerations 

The study protocol was approved by the Bioethics Committee of the Colegio Médicos de la Provincia de Salta, Argentina (N°14.200). Sera samples collected in each survey were treated according to the study protocols approved of each particular study. The study was conducted in accordance with principles of the 2013 Declaration of Helsinki. Participation was voluntary and informed consent was obtained from all patients involved in the study.

## 5. Conclusions

Our study confirms the presence of *S. stercoralis* infections at significant prevalence levels in the Chaco-Yungas regions of the Americas, highlighting the need for more in depth surveys within the region in order to understand the diversities within the region and address this and other public health needs of the impoverished communities living in this area. Integration of ivermectin to albendazole or mebendazole in MDA campaigns against STH would add coverage against *S. stercoralis* while also improving efficacy against *Trichuris trichiura*, therefore providing a tool that along with adequate sanitation and water facilities as well as health education, can aim for the control of STH in the Chaco-Yungas region. This study is also demonstrative of the advantages posed by serological methods in carrying larger scale surveys compared to stool methods that are less accurate, more difficult to standardize and integrate with other activities, and time consuming.

## Figures and Tables

**Figure 1 pathogens-09-00394-f001:**
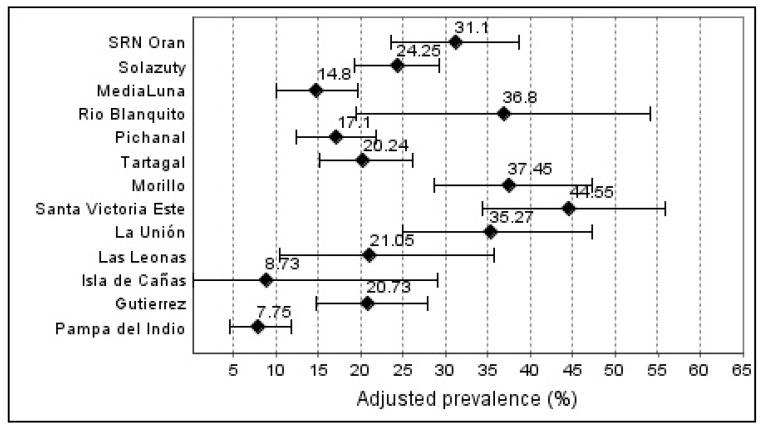
Seroprevalence of the *Strongyloides stercoralis* infection of localities from Chaco region Argentina-Bolivia.

**Figure 2 pathogens-09-00394-f002:**
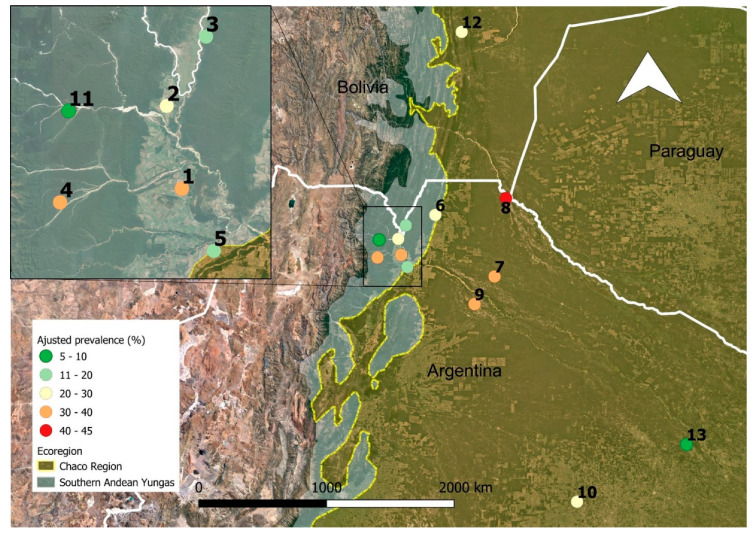
Distribution of seroprevalence by study site and ecoregions. Each number corresponds to a site, 1: SRN Oran; 2: Solazuty; 3: Medialuna; 4: Rio Blanquito; 5: Pichanal; 6: Tartagal; 7: Morillo; 8: Santa Victoria Este; 9: La Unión; 10: Las Leonas; 11: Isla de Cañas; 12: Gutierrez, and 13: Pampa del Indio.

**Figure 3 pathogens-09-00394-f003:**
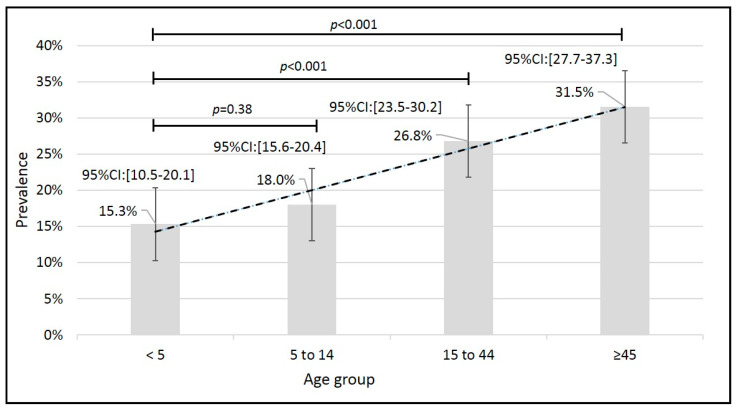
Age-specific prevalence of *Strongyloides stercoralis* infection by serologic result NIE-ELISA. All age groups were compared to children under five years old (age group < 5 years).

**Table 1 pathogens-09-00394-t001:** Characteristics of the population studied. masl: meters above sea level.

Localities	Department	Province	Country	Altitude (masl)	Region	Community
SRN Orán	Oran ^(1)^	Salta	Argentina	356m	Yunga	Periurban
Pichanal					Chaco	Urban
Solazuty					Yunga	Rural
Río Blanquito				773 m		
Isla de Cañas	Iruya ^(1)^			610 m		
Tartagal	General San Martín ^(1)^			495m	Chaco	Periurban-Rural
Medialuna					Yunga	Rural
La Unión	Rivadavia ^(1)^			227 m	Chaco	
Coronel Juan Solá (Morillo)				300 m		Urban-Rural
Santa Victoria Este				262 m		Rural
Pampa del Indio	Libertador Gral San Martin	Chaco		96 m	Chaco	
Las Leonas	12 de Octubre			124 m		
Gutierrez	Cordillera	Santa Cruz	Bolivia	800 m	Chaco	

^(1)^ Data from Anuario Estadístico 2016–2017. Dirección General de Estadística, provincia de Salta, Argentina.

**Table 2 pathogens-09-00394-t002:** Description of the different studies from which the samples included in this study were obtained.

Localities	Samples	Year of Study	Basic Sampling Unit	Population	Unimproved Water Supply	Unimproved Sanitation	Cited in
SRN Orán	142	2011	Household	Community	0%	0%	[14]
Pichanal	489	2013	Household	Community	0%	0%	[13]
Solazuty	182	2010	Household	Community	100%	100%	[14]
Río Blanquito	26	2010	Individuals	Community	100%	100%	None
Isla de Cañas	36	2011	Individual	Community	100%	100%	None
Tartagal	407	2012–2013	Household	Community	1%	82%	[31]
Medialuna	104	2016	Individual	Community	100%	100%	[27]
La Unión	128	2016	Individual	Community	6%	100%	None
Coronel Juan Solá (Morillo)	189	2016	Individual	Community	86%	100%	None
Santa Victoria Este	152	2016	Individual	Community	100%	100%	None
Pampa del Indio	585	2014	Individual	Community	100%	100%	None
Las Leonas	76	2008	Household	Community	100%	100%	[47]
Gutierrez	287	2011–2012	Household	Community	100%	100%	[43]

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
