# Peer review of "Seroprevalence of the Strongyloides stercoralis Infection in Humans from Yungas Rainforest and Gran Chaco Region from Argentina and Bolivia"

_pathogens, 2020, doi:10.3390/pathogens9050394_

Round 1

Reviewer 1 Report

The paper presents monitoring study concerning human Strongyloides stercoralis infection in south America in Gran Chaco. There are relatively wide study covering more than 2800 people from 13 carried out during six years.

The strong side on the study is the use of modern serological method based on recombinant antigen of S. stercoralis, characterized by relatively high specificity and sensitivity. Results were calculated and presented in clear way with proper statistic tests. (Well presented on charts). Interesting correlation of infection with age was confirmed. Discussion was led in accessible way comfortable for readers.

What can be added: in my opinion more information about life cycle of parasite should be added to introduction. Especially about ways of infection underlining the fact that larvae can penetrate the skin when it contact soil. And also about autoinfection phenomenon (in which larvae become infective in intestine … etc.). it delights a course of infection and the case of relatively strong immunoresponse connected with the migration of larve through the human body.

Reviewer 2 Report

I have read with interest the manuscript sent by Dr Cimino et al. The authors present the results of a seroprevalence study of S. stercoralis infection conducted in 13 locations of Gran Chaco region and Yungas rainforest. The manuscript is well written, methodologically correct, and it addresses a topic of high interest. One of the main limitations for the people working on strongyloidiasis control is the lack of epidemiological information. I have some comments and suggestions:

  • Strongyloides stercoralis does not appear in italics repeatedly. Please, revise it throughout the manuscript.
  • Was the informed consent obtained from the study participants? If not, give reason why.
  • Could you explain more in depth how the study participants were selected? There is no information regarding the inclusion criteria.
  • Did the patients receive specific treatment in those cases where the serology was positive?
  • In the light of the obtained results, could the authors suggest any strategy for the stercoralis infection control in this area? Sanitation improvement? Mass drug administration campaigns?

Reviewer 3 Report

Dear Editor,

The manuscript entitled "Seroprevalence of the Strongyloides stercoralis infection in humans from yungas rainforest and Gran Chaco region, Argentinian-Bolivian" presents the results of a seroprevalence study of this nematode in 13 locations in these regions of Argentina and Bolivia during the period 2010-2016. The results show a global prevalence around 20%, with important differences between locations (7.75%-44.55%) and different age groups. These results show the need to carry out disease control measures on these impoverished communities. The topic is interesting and the study is of great value from a public health point of view. Although serology is not the standard gold test, Strongyloides serology has good sensitivity and allows the diagnosis of a greater number of infections than with direct methods. I find it interesting and useful for Pathogens readers

Broad comments

The language is correct, the bibliographic review is adequate and updated, and the extension is adapted to the requirements of the journal. The manuscript is correctly structured. The objective is well established. The methodology is correct. The results are presented clearly and systematically.

Specific comments

Below I show some comments from which the manuscript could benefit.

  • Review Strongyloides stercoralis throughout the text, it should be in italics.
  • In figure 2, authors indicate significant differences between 5-14 years-old group and ≥ 45 years-old one. Are there no differences between the group of <5 years-old and ≥ 45 years-old? And between the group of <5 years-old and the group of 15-44 years-old? Clarify this point.
  • Line 190, Universidad should be written correctly.
